# CAN MLLMs GUIDE ME HOME? A BENCHMARK STUDY ON FINE-GRAINED VISUAL REASONING FROM TRANSIT MAPS

## ABSTRACT

Multimodal large language models (MLLMs) have recently achieved significant progress in visual tasks, including semantic scene understanding and text-image alignment, with reasoning variants enhancing performance on complex tasks involving mathematics and logic. However, their capacity for reasoning tasks involving fine-grained visual understanding remains insufficiently evaluated. To address this gap, we introduce REASONMAP, a benchmark designed to assess the fine-grained visual understanding and spatial reasoning abilities of MLLMs. REASONMAP encompasses high-resolution transit maps from 30 cities across 13 countries and includes 1,008 question-answer pairs spanning two question types and three templates. Furthermore, we design a two-level evaluation pipeline that properly assesses answer correctness and quality. Comprehensive evaluations of 15 popular MLLMs, including both base and reasoning variants, reveal a counterintuitive pattern: among open-source models, base models outperform reasoning ones, while the opposite trend is observed in closed-source models. Additionally, performance generally degrades when visual inputs are masked, indicating that while MLLMs can leverage prior knowledge to answer some questions, fine-grained visual reasoning tasks still require genuine visual perception for strong performance. Our benchmark study offers new insights into visual reasoning and contributes to investigating the gap between open-source and closed-source models.

## 1 INTRODUCTION

Multimodal large language models (MLLMs) (Achiam et al., 2023; Bai et al., 2025; Zhu et al., 2025; Hu et al., 2025; Li et al., 2025b) have recently achieved notable advancements across a range of vision-language tasks, including visual grounding (Peng et al., 2023; Yang et al., 2024b), reasoning segmentation (Chen et al., 2024; Zhang et al., 2024b; Ren et al., 2024; Lai et al., 2024; Wang et al., 2025a), and text-image alignment (Yue et al., 2025a; Yarom et al., 2023). Building upon these developments, reasoning MLLMs (OpenAI, 2024b; Guo et al., 2025a; Team et al., 2025; Wei et al., 2025; Peng et al., 2025; ByteDance, 2025; Qwen Team, 2024) have further improved performance on complex visual reasoning tasks such as visual math problems (Yang et al., 2024c; Wang et al., 2024a), visual question answering (VQA) (Shiri et al., 2024; Yue et al., 2024; Wang et al., 2024a), and spatial reasoning (Shiri et al., 2024; Li et al., 2025a; Dihan et al., 2024). These capabilities are critical for a wide range of real-world applications, including embodied AI, autonomous agents, and decision-making systems such as autonomous driving (Duan et al., 2022; Wang et al., 2024b; Cui et al., 2024). As multimodal tasks grow in complexity and practical relevance, the need for rigorous benchmarks to assess fine-grained visual reasoning becomes increasingly essential.

To address the growing demand for robust evaluation of multimodal reasoning, several benchmarks have been proposed. Datasets such as MathVQA (Wang et al., 2024a) and MMMU (Yue et al., 2024) incorporate multimodal questions but often permit models to succeed via shallow heuristics, without requiring genuine visual grounding. MathVerse (Zhang et al., 2024a) mitigates this limitation by introducing diverse problem variants that encourage reliance on visual input. VisuLogic (Xu et al., 2025b) further enforces visual reasoning by explicitly eliminating language-only shortcuts. Other efforts, such as VisualPuzzles (Song et al., 2025), VGRP-Bench (Ren et al., 2025), and R-Bench (Guo et al., 2025c), target logical and structural reasoning, while CityBench (Feng et al.,

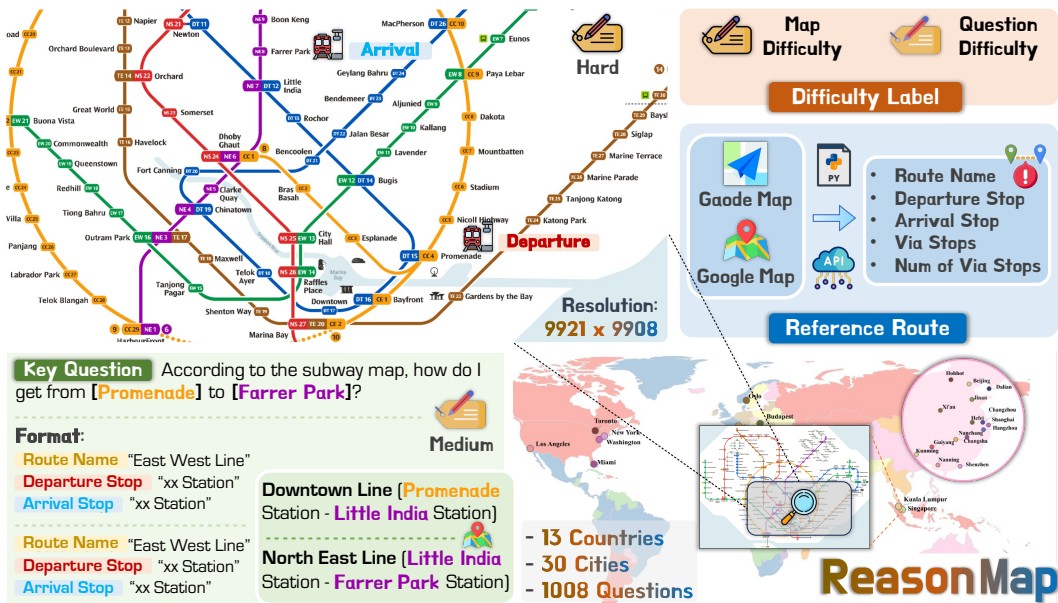

Figure 1: Overview of REASONMAP. We introduce a benchmark dataset designed to evaluate fine-grained visual reasoning abilities of MLLMs, encompassing 1,008 question-answer pairs constructed over high-resolution transit maps from 30 cities, spanning two question types and three templates.

2024) and DriveBench (Xie et al., 2025) focus on domain-specific applications like urban tasks and autonomous driving. V*Bench (Wu & Xie, 2024) emphasizes detailed visual understanding. MapBench (Xing et al., 2025) addresses spatial reasoning by introducing structured scene graphs for map navigation. Despite these advances, systematic evaluation of fine-grained visual reasoning remains limited, especially for structured and information-rich diagrams like high-resolution transit maps, leaving a critical gap in existing benchmarks.

In this paper, we introduce REASONMAP (Figure 1), a benchmark designed to evaluate the fine-grained visual understanding and spatial reasoning capabilities of MLLMs using high-resolution transit maps. As structured and information-dense visual artifacts, maps inherently require precise spatial interpretation, making them well-suited for assessing detailed visual reasoning. REASONMAP comprises 1,008 human-verified question-answer pairs spanning 30 cities across 13 countries. Each instance includes a map, two stops, two questions (*short* and *long*), multiple reference routes, and two difficulty labels (*map* and *question* difficulty). The questions cover two types and three prompting templates capturing both coarse and fine-grained spatial reasoning. To ensure data quality, we perform manual route verification, promote question diversity, and balance difficulty distribution. For evaluation, we propose a two-level framework that independently measures answer correctness (via *accuracy*) and quality (via a proposed *map score*), reflecting both feasibility and efficiency.

We conduct comprehensive experiments on 15 widely-used MLLMs, encompassing base and reasoning models. Our results reveal a counterintuitive finding: Among open-source models, base variants outperform their reasoning counterparts, whereas the opposite holds for closed-source models. Moreover, when only textual inputs are provided, models can still answer some questions based on inner knowledge, but in most cases, their performance noticeably drops. This highlights a critical limitation in the current model behavior. While some models can leverage prior knowledge and textual cues to solve certain tasks, the tasks (*e.g.*, fine-grained visual reasoning tasks) requiring genuine visual understanding still necessitate effective integration of multimodal information for robust reasoning.

Our main contributions are summarized as follows: (1) We develop an extensible, semi-automated pipeline for dataset construction, facilitating scalable expansion to additional maps and cities. Using this pipeline, REASONMAP is constructed to evaluate fine-grained visual reasoning capabilities in MLLMs; (2) We propose a structured two-level evaluation framework that separately quantifies answer correctness and quality using accuracy and the proposed map score, respectively, enabling fine-grained answer assessment; and (3) A comprehensive benchmarking study is conducted across 15 MLLMs, providing insights into model performance, robustness, and the interplay between visual and textual cues, thereby informing future research on multimodal reasoning.

Table 1: Comparison between REASONMAP and existing multimodal reasoning datasets. For entries in the dataset size column with notation like ($\times n$), each base problem has multiple versions to enforce visual grounding. Specifically, VGRP-Bench is constructed by sampling over 20 core puzzles.

| Name | Year | Dataset Size | Avg. Resolution | Training Set | Step Evaluation | Multilingual (Count) |
|---|---|---|---|---|---|---|
| MMMU (Yue et al., 2024) | 2024 | 11.5k | $684 \times 246$ | ✗ | ✗ | ✓ (2) |
| MathVerse (Zhang et al., 2024a) | 2024 | $2,612$ ($\times 6$) | $577 \times 487$ | ✗ | ✗ | ✗ |
| VisuLogic (Xu et al., 2025b) | 2025 | $1,003$ | $601 \times 331$ | ✓ | ✗ | ✗ |
| VisualPuzzles (Song et al., 2025) | 2025 | $1,168$ | $767 \times 464$ | ✗ | ✗ | ✗ |
| VGRP-Bench (Ren et al., 2025) | 2025 | 20 ($\times 5$) | $790 \times 790$ | ✗ | ✓ | ✗ |
| R-Bench (Guo et al., 2025c) | 2025 | 665 | $629 \times 348$ | ✗ | ✗ | ✓ (2) |
| V*Bench (Wu & Xie, 2024) | 2023 | 191 | $2,246 \times 1,582$ | ✗ | ✗ | ✗ |
| REASONMAP | 2025 | $1,008$ ($\times 2$) | $5,839 \times 5,449$ | ✓ | ✓ | ✓ (4) |

## 2 RELATED WORK

**Reasoning in LLMs & MLLMs.** Recent advances in large language models (LLMs) have demonstrated significant improvements in reasoning capabilities through reinforcement fine-tuning paradigms (OpenAI, 2024b; Guo et al., 2025a; Feng et al., 2025; Hendrycks et al., 2021), which leverage GRPO (Shao et al., 2024) to unlock the reasoning potential of LLMs. This paradigm has also been extended to the multimodal domain, with increasing interest in applying reinforcement learning (RL) to visual reasoning (Team, 2025; Lab, 2025; Liu et al., 2025; Tan et al., 2025; Shen et al., 2025). Both open-source and closed-source communities have introduced advanced reasoning MLLMs built upon earlier systems (Zhu et al., 2025; Bai et al., 2025; Yang et al., 2024a; OpenAI, 2025). Notable open-source models include Kimi-VL (Team et al., 2025), Skywork-R1V (Wei et al., 2025; Peng et al., 2025), and Qwen-QvQ (Qwen Team, 2024), whereas Doubao-1.5-Pro (ByteDance, 2025), Seed1.5-VL (Guo et al., 2025b), OpenAI o3 (OpenAI, 2025), OpenAI 4o (OpenAI, 2024a), and Gemini (Gemini et al., 2023) represent leading closed-source efforts. Despite recent progress, systematic evaluation of fine-grained visual reasoning in MLLMs still remains limited, as existing benchmarks primarily target coarse-grained tasks and fail to capture model performance on complex real-world visual content.

**Multimodal Reasoning Datasets.** As multimodal reasoning has rapidly progressed, various benchmarks have emerged to evaluate MLLMs across different reasoning dimensions (see summary in Table 1). Datasets such as V*Bench (Wu & Xie, 2024), VisualPuzzles (Song et al., 2025), VisuLogic (Xu et al., 2025b), and VGRP-Bench (Ren et al., 2025) primarily examine abstract visual reasoning through synthetic tasks involving logic, structure, and pattern recognition. In parallel, CityBench (Feng et al., 2024) and DriveBench (Xie et al., 2025) shift focus to real-world spatial reasoning, assessing model performance on complex urban or autonomous driving scenarios. For mathematical reasoning, benchmarks like MathVQA (Wang et al., 2024a), MMMU (Yue et al., 2024), and MathVerse (Zhang et al., 2024a) integrate multimodal inputs, with MathVerse notably introducing varied problem formats to strengthen visual dependence. Additionally, MapBench (Xing et al., 2025) employs structured scene graphs combined with CoT prompting to support navigation tasks based on manually curated and verified questions. Its image resolution, while relatively low, reflects a common characteristic shared by current datasets. Unlike these works, we first introduce a benchmark for evaluating fine-grained visual reasoning capacities with high-resolution transit maps.

**Map-based Spatial Reasoning.** Among the many directions of multimodal reasoning, map-based spatial reasoning has emerged as a crucial area, with broad applications in navigation, urban planning, and autonomous systems (Bao et al., 2023; Seff & Xiao, 2016; Xu et al., 2025a; Wang et al., 2023). Recent efforts have focused on enabling models to interpret and reason over various types of map data. CityBench (Feng et al., 2024) provides a dataset for evaluating urban scene understanding, while MapLM (Cao et al., 2024) introduces a benchmark for map and traffic scene understanding. PlanAgent (Zheng et al., 2024) and PReP (Zeng et al., 2024) explore embodied planning in environments that require interpreting map information. MapEval (Dihan et al., 2024) proposes a structured evaluation suite for map reasoning, and GeoNav (Xu et al., 2025a) investigates geospatial navigation using LLMs. Most existing methods (Dihan et al., 2024; Feng et al., 2024; Zheng et al., 2024) depend on external tools (*e.g.*, map services or APIs) to complete spatial tasks, which often bypasses the need for genuine visual reasoning. However, spatial reasoning based on visual understanding remains essential. Our work aims to fill this gap by evaluating such capabilities without tool assistance.

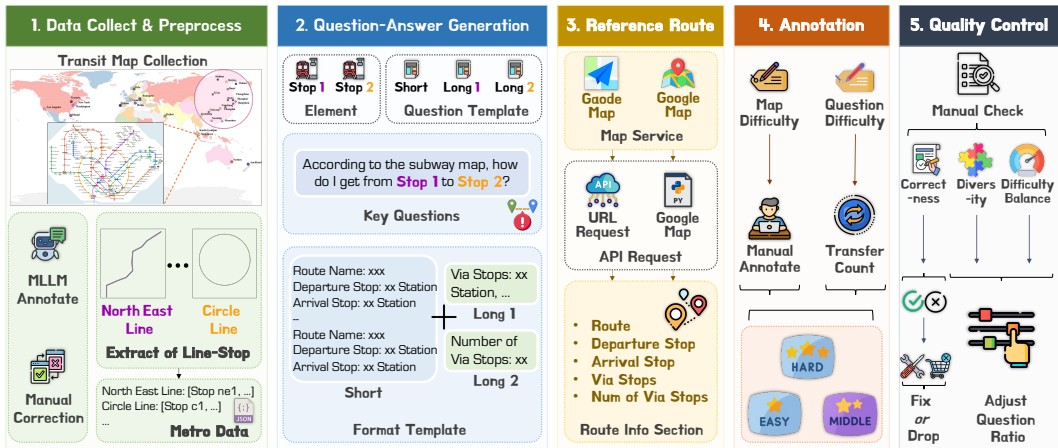

Figure 2: The building pipeline of REASONMAP consists of **three** main stages: **(1)** data collection and preprocessing, **(2)** question–answer pair construction, and **(3)** quality control. Steps (2-4) in the figure correspond to the question–answer pair construction stage. Zoomed-in for more details.

## 3 REASONMAP CONSTRUCTION

In this section, we first present the complete dataset building pipeline as shown in Figure 2, which consists of three main stages: (1) data collection and preprocessing, (2) question–answer pair construction, and (3) quality control. We then report comprehensive statistics of the dataset.

### 3.1 REASONMAP BUILDING PIPELINE

#### 3.1.1 DATA COLLECTION AND PREPROCESSING

We collect and manually select 30 high-resolution transit maps covering 30 cities across 13 countries from publicly available online sources, in compliance with relevant licenses and regulations, ensuring diversity and a balanced range of map difficulty. We then leveraged MLLMs to extract the names of transit lines and their corresponding stops, followed by manual correction to ensure correctness. Special cases such as transfer stops and branch-starting stops were annotated in a standardized format appended to the respective stop entries. Finally, for subsequent usage, all route and stop information was saved in a unified JSON format, referred to as the Metro Data.

#### 3.1.2 QUESTION-ANSWER PAIR CONSTRUCTION

The construction of question–answer pairs involves three key steps: (1) Question Generation, where we formulate questions based on predefined templates; (2) Reference Route Collection, where we obtain corresponding reference routes using Gaode Map[1] and Google Map[2]; and (3) Label Annotation, where we assign difficulty levels for both the maps and the questions.

**Question Generation.** We randomly select two stops (refer to $stop_1$ and $stop_2$) from the current high-resolution transit map. We then generate one short question and one long question based on predefined question templates and two stops (Figure 2). The short question has only one fixed template, while the long question is randomly assigned one of two available templates during generation. Additionally, the two long question templates differ in focus: one asks for the number of via stops, while the other requires identifying each via stop (see detailed templates in Appendix A.1).

**Reference Route Collection.** For each question, we query all valid transit routes between $stop_1$ and $stop_2$ using APIs from map services (*e.g.*, Gaode Map for Chinese cities and Google Map for other cities). The retrieved routes are stored in a unified format containing relevant metadata (*e.g.*, route name, departure stop, arrival stop, via stops, and number of via stops). We discard routes that cannot be visually traced on the map, ensuring consistency with the visual content.

---

[1] https://console.amap.com/dev/index
[2] https://developers.google.com/maps/apis-by-platform

**Label Annotation.** Two levels of difficulty labeling are included in this stage. For map difficulty, we manually assign each map to one of three difficulty levels (easy, medium, hard), ensuring a balanced split across 30 maps, with 10 maps per level. For question difficulty, we assign difficulty based on the number of transfers in the reference route: routes with no transfers are labeled as easy, those with one transfer as medium, and all others as hard. To ensure balance, we set a fixed difficulty distribution threshold of $20 : 15 : 5$ (easy:medium:hard) for each map, generating 40 questions. Once the quota for a difficulty level is reached on a given map, no additional questions of that level are retained. Additionally, we provide a more fine-grained taxonomy of questions as a reference in Appendix A.2.

### 3.1.3 Quality Control

To ensure the reliability and balance of the dataset, we perform quality control from three perspectives: correctness, diversity, and difficulty balance. Incorrect question–answer pairs are either manually corrected or discarded. We then involve both automatic checks and manual adjustments to ensure consistency and coverage across all difficulty levels. One reserved example is shown in Figure 1.

### 3.2 Dataset Statistics

The REASONMAP consists of 30 high-resolution transit map images (see map sources in Appendix A.3) with an average resolution of $5,839 \times 5,449$ pixels. In total, it contains $1,008$ question–answer pairs, including stop names in four languages (*e.g.*, English, Hungarian, Chinese, and Italian). The distribution of question difficulty is as follows: $57.7\%$ are labeled as easy, $34.4\%$ as medium, and $7.8\%$ as hard. Additionally, a subset of 312 samples is manually selected as the test set for the benchmark experiments described in Section 5, while the remaining samples serve as a training set for future use. To ensure diversity and difficulty balance, the test set includes 11 cities with a $4 : 3 : 4$ map difficulty ratio and a question difficulty distribution (181 easy, 108 medium, 23 hard) that maintains consistency with the full dataset. Moreover, REASONMAP includes inter-modal transfers in cities like Sydney, where subways, light rail, and airport lines converge.

## 4 Evaluation Framework

This section systematically introduces a two-level evaluation framework for assessing model performance on the REASONMAP. This framework separately evaluates the correctness and quality of answers produced by models. Specifically, we quantify correctness using accuracy and design map score to measure the quality of answers, considering multiple factors (*e.g.*, route efficiency and alignment with the reference routes from map services).

**Preparation for Evaluation.** We first parse the model-generated answers according to the required format. Answers that do not comply with the specified format or cannot be parsed due to model hallucination (Bai et al., 2024) are marked as invalid. Invalid responses are excluded from subsequent evaluations, with accuracy and map score set to zero. For the correctness evaluation, we utilize the Metro Data mentioned in Section 3.1.1 as ground truth. For the quality evaluation, we adopt the collected reference routes as presented in Section 3.1.2 as the ground truth.

### 4.1 Correctness Evaluation

We evaluate the correctness of the answer using Algorithm 1 in Appendix B. Specifically, the evaluation checks the correctness of the overall departure and arrival stops ($stop_1$ and $stop_2$), verifies if the route name of each segment exists in the Metro Data, ensures the departure and arrival stops are valid for each segment, and confirms that transfer stops between consecutive segments are consistent. An answer is considered correct only if all the above checks are satisfied. Additionally, we apply the same correctness evaluation algorithm to the answers of short and long questions.

### 4.2 Quality Evaluation

To evaluate the quality of the answers, we introduce a unified scoring metric, referred to as the map score, which is applied to both short and long questions using the evaluation procedure (see Algorithm 2 in Appendix B). The overall evaluation framework for route quality follows a structure

Table 2: Evaluations of various MLLMs on REASONMAP. $S.$ represents results for short questions, while $L.$ denotes results for long questions. The map score is capped at 20 for short questions, while for long questions, the maximum score is 40. **Bold** indicates the best results among open-source and closed-source models, respectively, while underline represents the second best. We report more fine-grained error analysis metrics in Table A4.

| Model | Type | Weighted Acc. ($S.$) | #Tokens ($S.$) | Weighted Acc. ($L.$) | #Tokens ($L.$) | Weighted Map Score ($S. / L.$) |
|---|---|---|---|---|---|---|
| *Open-source Models* | | | | | | |
| Qwen2.5-VL-3B-Instruct (Bai et al., 2025) | Base | 8.68% | 42 | 7.99% | 151 | 2.75 / 3.70 |
| Qwen2.5-VL-32B-Instruct (Bai et al., 2025) | Base | 16.49% | 36 | 15.71% | 112 | 3.88 / 6.84 |
| Qwen2.5-VL-72B-Instruct (Bai et al., 2025) | Base | **26.65%** | 33 | **24.22%** | 104 | **5.09 / 8.80** |
| InternVL3-38B (Zhu et al., 2025) | Base | 14.84% | 43 | 13.45% | 68 | 3.48 / 6.31 |
| InternVL3-78B (Zhu et al., 2025) | Base | 25.35% | 33 | 19.62% | 62 | 4.80 / 7.50 |
| Kimi-VL-A3B-Instruct (Team et al., 2025) | Base | 12.76% | 41 | 12.33% | 41 | 3.30 / 5.37 |
| Kimi-VL-A3B-Thinking (Team et al., 2025) | Reasoning | 5.47% | 754 | 5.47% | 1,287 | 2.44 / 3.17 |
| Skywork-R1V-38B (Wei et al., 2025; Peng et al., 2025) | Reasoning | 6.86% | 645 | 3.21% | 842 | 2.11 / 3.11 |
| QvQ-72B-Preview (Qwen Team, 2024) | Reasoning | 9.03% | 1,279 | 4.25% | 1,619 | 1.59 / 1.55 |
| *Closed-source Models* | | | | | | |
| Doubao-115 (ByteDance, 2025) | Base | 34.20% | 32 | 38.02% | 118 | 5.25 / 11.96 |
| OpenAI 4o (OpenAI, 2024a) | Base | 41.15% | 34 | 42.80% | 58 | 6.84 / 13.57 |
| Doubao-415 (ByteDance, 2025) | Reasoning | 43.14% | 536 | 46.09% | 1,796 | 7.33 / 14.67 |
| Doubao-428 (ByteDance, 2025) | Reasoning | 37.15% | 532 | 37.85% | 2,167 | 5.52 / 11.73 |
| Gemini-2.5-Flash (Gemini et al., 2023) | Reasoning | 46.09% | 806 | 29.86% | 1,419 | 7.64 / 9.98 |
| OpenAI o3 (OpenAI, 2025) | Reasoning | **63.02%** | 1,236 | **59.11%** | 2,372 | **9.53 / 17.96** |

similar to that used in Section 4.1. The following evaluation procedure assumes a single reference route for simplicity. In practice, if multiple reference routes are available, the answer is evaluated against each of them, and the highest score is taken as the final map score.

For short questions, the map score solely focuses on route-level and endpoint consistency, excluding all long-question-specific parts. We compute the score by comparing segment pairs in the answer and reference route. Specifically, correctly matching $stop_1$ and $stop_2$ contributes one point, matching the route name adds two points, and matching the departure and arrival stops within each route segment provides one point each. The final score is capped at 10, and an additional bonus is awarded if the answer is judged correct based on the correctness evaluation procedure described in Section 4.1. This design ensures that a correct answer always receives a higher score than any incorrect one.

For long questions, the evaluation incorporates additional scoring components tailored to the two question templates introduced in Section 3.1.2. These components are designed to capture the increased reasoning depth required in long-form responses. As with short questions, a bonus score is also added for correct answers. The two additional scoring components are detailed below.

**Via Stop Count Evaluation.** For long questions that require models to predict the number of via stops for each segment, we introduce the *num_via_stop_score*. This score compares the via stop count of the answer and reference route by computing the absolute error and mapping it to a fixed score (4). A perfect match yields full points, while larger discrepancies receive proportionally lower scores. The score is then capped at 10 for the full route.

**Specific Via Stop Evaluation.** For long questions that require explicit enumeration of intermediate stops, we compute *via_stop_score* using a combination of two factors: the number of correctly matched via stops, and the intersection-over-union (IoU) between via stop sets of the answer and reference route. The final score for this component is obtained by averaging the IoU score (scaled to 10) and the exact match count (capped at 10), and then clipped to a maximum of 10.

## 5 EXPERIMENTS

### 5.1 EXPERIMENTAL SETUPS

We conduct extensive benchmark experiments on REASONMAP using 15 popular MLLMs under different inference settings, analyzing their performance and comparing results. Several interesting insights emerge from this comparison. The detailed experimental settings are described below.

**Evaluated Models.** We evaluate a diverse set of MLLMs categorized into two groups based on whether they are reasoning-oriented models with a long-thinking process. Reasoning models include: Skywork-R1V-38B (Wei et al., 2025; Peng et al., 2025), QvQ-72B-Preview (Qwen Team, 2024), Kimi-VL-A3B-Thinking/Instruct (Team et al., 2025), OpenAI o3 (OpenAI, 2025), Gemini-2.5-

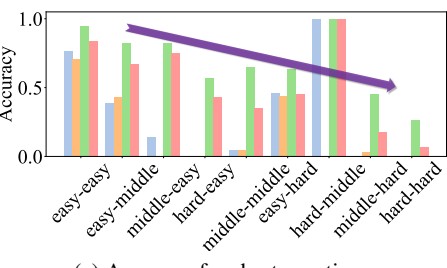 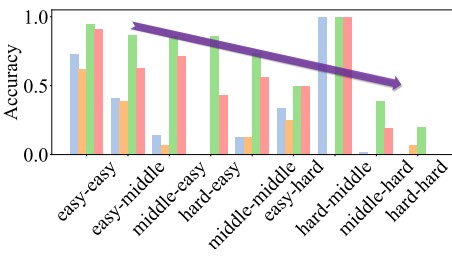

(a) Accuracy for short questions  (b) Accuracy for long questions

Figure 3: Accuracy across difficulty combinations for four representative MLLMs (Qwen2.5-VL-72B-I, InternVL3-78B, OpenAI o3, and Doubao-415). Each difficulty combination is denoted by a pair (*e.g.*, *easy-hard*), where the first term indicates question difficulty and the second term represents map difficulty. The pair (*hard-middle*) contains only one sample, leading to an accuracy of 100%.

Flash (Gemini et al., 2023), Doubao-1-5-thinking-vision-pro-250428 (Doubao-428), and Doubao-1.5-Thinking-Pro-M-250415 (Doubao-415) (ByteDance, 2025). Base models include: Qwen2.5-VL series (3B, 32B, 72B) (Bai et al., 2025), InternVL3 series (38B, 78B) (Zhu et al., 2025), OpenAI 4o (OpenAI, 2024a), and Doubao-1.5-Vision-Pro-32k-250115 (Doubao-115) (ByteDance, 2025). Additionally, the Doubao 1.5 Pro series has an activated parameter size of 20B.

**Inference Settings.** For open-source models, we set the max output token limit to $2,048$, while keeping other parameters consistent with the official HuggingFace configurations. All open-source models are deployed using PyTorch and the HuggingFace Transformers library on 8 NVIDIA A100 GPUs. For closed-source models, we access their official APIs for evaluation and follow the default settings provided by each model's official documentation. We further discuss the diverse image processing strategies when handling high-resolution visual inputs in Appendix B.2.

**Difficulty-Aware Weighting.** To better reflect the varying complexity of different samples, we adopt a difficulty-aware weighting strategy based on the combination of question difficulty and map difficulty. Specifically, each difficulty pair is assigned a predefined weight, with harder combinations receiving higher values. The complete weight matrix is provided in Appendix B.3. Both accuracy and map score are evaluated using this weighting scheme, ensuring that models are more strongly rewarded for correctly solving more challenging examples.

## 5.2 EXPERIMENTAL RESULTS

### 5.2.1 PERFORMANCE OF MLLMS WITH FULL INPUT

The principal results are summarized in Table 2. Notably, we observe a counterintuitive phenomenon: among open-source models, reasoning models consistently underperform their base counterparts, whereas the opposite holds in the closed-source setting[3]. Prior work suggests that RL may improve sample efficiency without introducing fundamentally new reasoning abilities (Yue et al., 2025b; Wang et al., 2025b; Zhang et al., 2025), while RL-trained models tend to bias their output distributions toward high-reward trajectories, which helps produce more correct responses but may simultaneously constrain the model's exploration capacity and reduce its ability to leverage broader foundational knowledge. In addition, recent studies indicate that multimodal models may sometimes rely on inner knowledge priors instead of truly attending to visual inputs (Jiang et al., 2024; Hao et al., 2025; Ghatkesar et al., 2025; Zhang et al., 2024a). This tendency is further supported by the results in Section 5.2.2, where open-source models still maintain part of their performance even without visual input, indicating limited visual grounding. In contrast, closed-source reasoning models outperform their base variants. One possible explanation lies in the broader knowledge coverage and better visual integration observed in these models (ByteDance, 2025; OpenAI, 2025; Gemini et al., 2023).

We further analyze the effect of model size by examining performance within the same architecture series. Qwen2.5-VL and InternVL series show a consistent trend: larger models achieve better accuracy with fewer tokens, suggesting that the scaling law (Kaplan et al., 2020) continues to hold even in fine-grained visual reasoning tasks. Figure 3 presents accuracy distributions across different

---

[3]Although the comparison across closed-source models may not be fair due to lack of transparency in details, the reasoning variants exhibit stronger performance in this category.

Table 3: Evaluations of MLLMs on REASONMAP w/o visual inputs. **S**. denotes results for short questions and **L**. denotes results for long questions. The map score is capped at 20 for short questions, while for long questions, the maximum score is 40. **Bold** indicates the best results among open-source and closed-source models, respectively, while underline represents the second best. Green highlights improved results compared to the full input setting (Table 2), while red indicates performance drops.

| Model | Type | Weighted Acc. (S.) | #Tokens (S.) | Weighted Acc. (L.) | #Tokens (L.) | Weighted Map Score (S. / L.) |
|---|---|---|---|---|---|---|
| *Open-source Models* | | | | | | |
| Qwen2.5-VL-3B-Instruct (Bai et al., 2025) | Base | 9.38%↑0.7% | 47 | 9.72%↑1.73% | 147 | 2.93↑0.18 / 4.51↑0.81 |
| Qwen2.5-VL-72B-Instruct (Bai et al., 2025) | Base | **16.41%**↓10.24% | 28 | **15.71%**↓8.51% | 108 | **4.03**↓1.06 / **6.49**↓2.31 |
| Kimi-VL-A3B-Instruct (Team et al., 2025) | Base | 11.81%↓0.95% | 41 | 9.81%↓2.52% | 49 | 3.37↑0.07 / 5.32↓0.05 |
| Kimi-VL-A3B-Thinking (Team et al., 2025) | Reasoning | 4.17%↓1.30% | 1,039 | 2.08%↓3.39% | 1,755 | 2.06↓0.38 / 1.64↓1.53 |
| *Closed-source Models* | | | | | | |
| Doubao-115 (ByteDance, 2025) | Base | 13.72%↓20.48% | 34 | 13.98%↓24.04% | 99 | 3.50↓1.75 / 6.48↓5.48 |
| Doubao-415 (ByteDance, 2025) | Reasoning | **21.53%**↓21.61% | 352 | **17.19%**↓28.90% | 1,047 | **4.85**↓2.48 / **7.68**↓6.99 |

combinations of question and map difficulty. As expected, performance degrades as task complexity increases. Additionally, Figure 4 illustrates accuracy variation across cities. We observe a negative correlation between map difficulty and accuracy. Moreover, model performance varies notably even among cities with comparable map difficulty levels. This disparity can be partially attributed to factors such as city prominence and the language used for stop names (see the ablation study results on language in Appendix C.3), both of which are closely tied to the model's pretrained knowledge. For instance, OpenAI o3 performs significantly better on complex cities like Singapore compared to Hangzhou, likely due to Singapore's higher international visibility and the use of English stop names, whereas Hangzhou is less prominent and its stop names are Chinese.

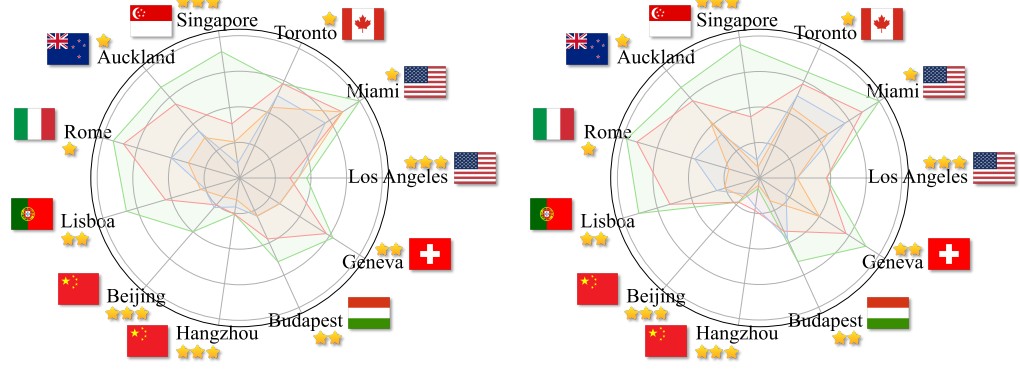

(a) Accuracy for short questions        (b) Accuracy for long questions

Figure 4: Accuracy across different cities for four representative MLLMs (Qwen2.5-VL-72B-I, InternVL3-78B, OpenAI o3, and Doubao-415). Each city is marked with the corresponding map difficulty and the country flag.

### 5.2.2 PERFORMANCE OF MLLMs WITHOUT VISUAL INPUT

To further investigate the reliance of MLLMs on visual input, we selected representative open-source and closed-source models for additional experiments, where the visual input was masked. The results are reported in Table 3. We observe that while most models can leverage prior knowledge to answer questions, their performance generally declines to varying degrees when visual input is removed, with the decline being more pronounced among closed-source models. Model performance is positively correlated with the performance drop after masking visual inputs, indicating effective use of visual information. In contrast, models like Qwen2.5-VL-3B-I show minimal or even improved performance, suggesting a reliance on prior knowledge rather than real visual reasoning. We further conduct non-vision experiments by replacing maps with their symbolic representations in Appendix C.4.

### 5.3 ERROR ANALYSIS

Figure 5 presents representative failure cases from REASONMAP, revealing several recurring error types. A common issue is *visual confusion*, where the model misidentifies the transit line due to similar colors or adjacent layouts, for instance, mistaking Line 9 for Line 16 (OpenAI o3, left column;

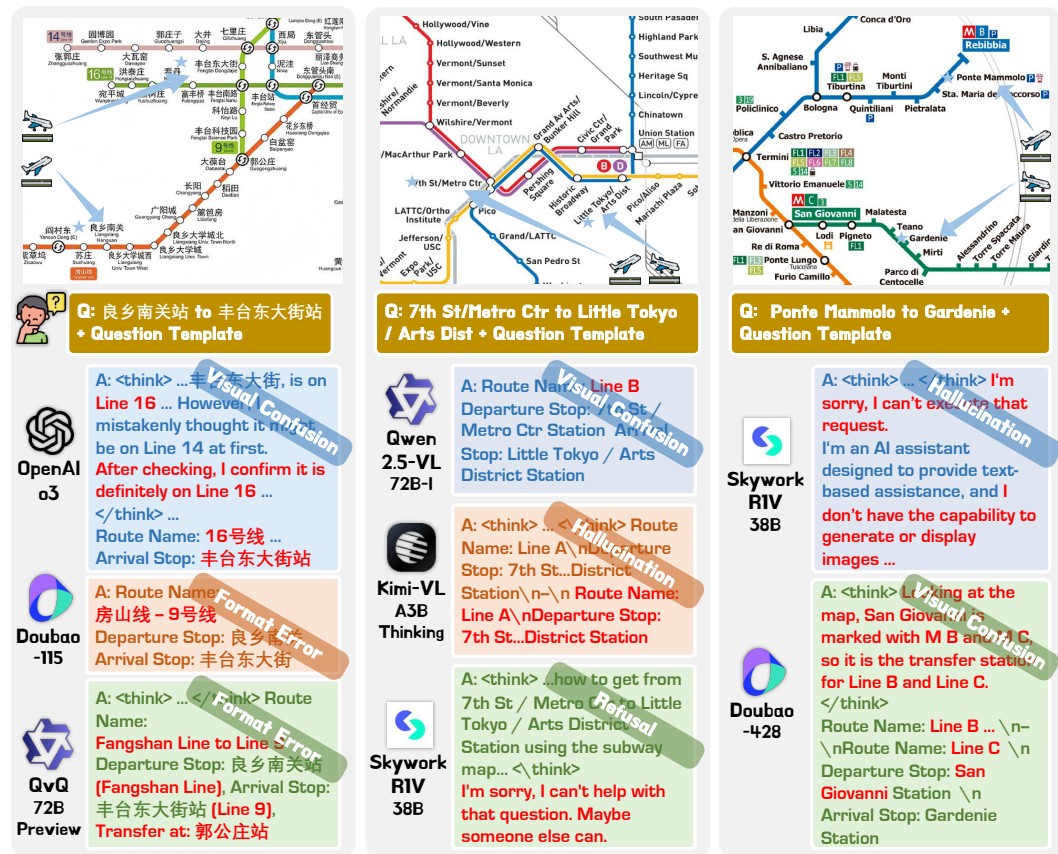

Figure 5: **Error case analyses** of various MLLMs using REASONMAP. For reasoning models, the reasoning process is explicitly marked with <think> and </think> tags. We highlight error contents in the answers with red and categorize them accordingly.

Doubao-428, right column). Another frequent problem is *format errors*, where responses deviate from the required structure, making them unprocessable despite containing correct route information (Doubao-115 and QvQ-72B-Preview, left column). We also observe instances of *hallucination* (Bai et al., 2024), where the model repeats the correct answer (Kimi-VL-A3B-Thinking, middle column) or generates information that is not present in the input, such as mentioning image generation, as seen in Skywork-R1V-38B (right column). *Refusal* cases are also present, where models explicitly decline to answer (Skywork-R1V-38B, middle and right column). Notably, these errors may occasionally co-occur within a single response (Skywork-R1V-38B, right column). Furthermore, we conduct a systematic analysis of failure causes from a model capability perspective (*e.g.*, optical character recognition (OCR), visual grounding, and spatial reasoning) in Appendix C.5. The above error types highlight the limitations in visual grounding and response robustness, especially when handling fine-grained visual details (see more case analyses in Appendix D).

## 6 CONLUSION

In this work, we introduce REASONMAP, a benchmark designed to evaluate the fine-grained visual understanding and spatial reasoning capabilities of MLLMs using high-resolution transit maps. Through a semi-automated and scalable data building pipeline, we curate a diverse set of human-verified question-answer pairs across 30 cities from 13 countries. Our two-level evaluation framework enables a nuanced assessment of both correctness and quality. Experimental results on 15 popular MLLMs reveal key insights into model behavior, highlighting performance gaps between base and reasoning models, as well as the crucial role of visual input. Error analyses further reveal recurring failure patterns (*e.g.*, visual confusion), highlighting weaknesses of current MLLMs in visual understanding and spatial reasoning. These findings underscore the need for more rigorous evaluation and training approaches to advance visual reasoning in multimodal systems.

## ETHICS STATEMENT

All experiments are conducted on REASONMAP, which is built using publicly available transit maps collected in compliance with relevant licenses and usage terms. The maps are selected to ensure geographic diversity and legal validity. Upon code release, we provide the source of each map for further reference. REASONMAP is intended solely for academic research on fine-grained visual understanding and spatial reasoning in MLLMs. It does not redistribute any copyrighted map images. All annotations are based on public information, contain no personal data, and are created under academic oversight. The benchmark is not intended for safety-critical use. We take care to ensure fairness, legal compliance, and responsible data handling. Additionally, we use the MIT License for code release on GitHub and the Apache License 2.0 for REASONMAP release on HuggingFace.

## REPRODUCIBILITY STATEMENT

To ensure reproducibility, we present evaluation setup details (*e.g.*, hardware and implementation) in Section 5.1 and Appendix B.3, and provide public implementation links in Appendix F.1 to facilitate rapid replication. We additionally release standardized splits and end-to-end instructions to reproduce all reported results (both code[4] and dataset[5]). During the review process, all links are anonymized and provided as supplements; upon acceptance, they will be replaced with permanent public links.

---

[4] https://github.com/Reason-Map/ReasonMap
[5] https://huggingface.co/datasets/AnonymousReasonMap/ReasonMap

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

APPENDIX

We provide a comprehensive overview in the Appendix, covering key details of our dataset, methodology, evaluation, analysis, and further discussions. Specifically, we include the question templates, a fine-grained taxonomy of difficulty, and sources of transit maps from 30 cities for REASONMAP construction in Appendix A. We then report detailed descriptions of the evaluation algorithm and experimental setup in Appendix B. In Appendix C, we conduct more exploratory experiments, including further RL training with training data in REASONMAP, evaluation of symbolic representation, and an ablation study about languages. We also provide the results of fine-grained error analysis metrics and systematically analyze failure causes. In Appendix D, we further extend case analysis by providing more classical cases. In addition, we further discuss the stated limitations, future directions, and potential broader impacts of our work in Appendix E. We finally present public implementation for the MLLMs used in our experiments and the statement of LLM usage (see Appendix F).

## A  DATASET CONSTRUCTION DETAILS

### A.1  QUESTION TEMPLATE SUMMARY

We present one short question template and two long question templates as follows.

**Short Question Template**

According to the subway map, how do I get from `[Stop 1]` to `[Stop 2]`? Provide only one optimal route, with only the line name and the departure and arrival stations. The format should be strictly followed:

```
Route Name: Line x
Departure Stop: xx Station
Arrival Stop: xx Station
--
Route Name: Line x
Departure Stop: xx Station
Arrival Stop: xx Station
```

**Long Question Template 1**

According to the subway map, how do I get from `[Stop 1]` to `[Stop 2]`? Provide only one optimal route, and include the number of via stops for each route section (excluding the departure and arrival stops). The format should be strictly followed:

```
Route Name: Line x
Departure Stop: xx Station
Arrival Stop: xx Station
Number of Via Stops: x
--
Route Name: Line x
Departure Stop: xx Station
Arrival Stop: xx Station
Number of Via Stops: x
```

**Long Question Template 2**

According to the subway map, how do I get from `[Stop 1]` to `[Stop 2]`? Provide only one optimal route, including all the via stops. The format should be strictly followed:

```
Route Name: Line x
Departure Stop: xx Station
Arrival Stop: xx Station
Via Stops: xx Station, xx Station, xx Station
--
Route Name: Line x
Departure Stop: xx Station
Arrival Stop: xx Station
Via Stops: xx Station
```

## A.2    A More Fine-grained Taxonomy of Difficulty

Beyond the easy, middle, and hard categorization for map and question difficulty, we provide three additional difficulty aware labels: 1) $city\_line\_count$, the total number of lines in a city (*i.e.*, a proxy for map difficulty); 2) $city\_transfer\_count$, the total number of transfer stations in a city (*i.e.*, a proxy for map difficulty); and 3) $question\_transfer\_count$, the number of transfers in the queried route (*i.e.*, a proxy for question difficulty). These labels enable fine-grained category design and filtering in subsequent analyses.

## A.3    Map Source

We provide the sources of all maps included in REASONMAP for further reference (Table A1).

Table A1: Source links for city transit maps used in REASONMAP. We present a total of 30 cities sourced from 13 countries.

| City | Source | City | Source | City | Source |
|---|---|---|---|---|---|
| Budapest | [Link] | Oslo | [Link] | Rome | [Link] |
| Lisboa | [Link] | Geneva | [Link] | Dubai | [Link] |
| Auckland | [Link] | Sydney | [Link] | Singapore | [Link] |
| Kuala Lumpur | [Link] | Los Angeles | [Link] | Miami | [Link] |
| New York | [Link] | Toronto | [Link] | Washington | [Link] |
| Guiyang | [Link] | Shanghai | [Link] | Huhehaote (Hohhot) | [Link] |
| Nanchang | [Link] | Nanning | [Link] | Shenzhen | [Link] |
| Hangzhou | [Link] | Dalian | [Link] | Kunming | [Link] |
| Hefei | [Link] | Beijing | [Link] | Changzhou | [Link] |
| Jinan | [Link] | Xi'an | [Link] | Changshang | [Link] |

## B EVALUATION DETAILS

### B.1 CORRECTNESS AND QUALITY EVALUATION

We present the detailed algorithms for evaluating answer correctness and quality in Section 4 (Algorithm 1 for correctness evaluation and Algorithm 2 for quality evaluation).

---

**Algorithm 1:** Correctness Evaluation

---

Initialize acc $\leftarrow$ 1;
**if** *departure stop of first segment $\neq$ stop$_1$ or arrival stop of last segment $\neq$ stop$_2$* **then**
  acc $\leftarrow$ 0;
**foreach** *segment in predicted route* **do**
  **if** *route name not in the Metro Data* **then**
    acc $\leftarrow$ 0;
  **if** *departure or arrival stop not in the stop list of the route* **then**
    acc $\leftarrow$ 0;
  **if** *not the last segment* **then**
    **if** *arrival stop of current segment $\neq$ departure stop of next segment* **then**
      acc $\leftarrow$ 0;

**return** acc

---

### B.2 HIGH-RESOLUTION IMAGE PREPROCESSING.

We compare how different Multimodal Large Language Models (MLLMs) handle high-resolution image inputs in Table A2. Specifically, we examine three key components in their preprocessing pipelines: dynamic resolution handling, positional encoding, and token compression.

1. **Dynamic resolution handling** refers to whether the model can directly accept images of arbitrary sizes without resizing or cropping. Most recent models support native resolution processing, enabling them to preserve fine-grained spatial information. In contrast, models like Gemini (Gemini et al., 2023) rely on image tiling and resizing to fit fixed input constraints.

2. **Positional encoding** helps the model retain spatial structure among visual tokens. Common strategies include 2D Rotary Positional Encoding (2D-RoPE) (Heo et al., 2024), as seen in Qwen2.5-VL (Bai et al., 2025) and Doubao (ByteDance, 2025), or flexible alternatives like V2PE (Ge et al., 2024) in InternVL3 (Zhu et al., 2025). Some models (*e.g.*, Gemini, Skywork-R1V (Wei et al., 2025; Peng et al., 2025)) do not explicitly disclose their positional encoding scheme, which we mark as "–" in the table.

3. **Token compression** aims to reduce the number of visual tokens for more efficient processing. Different models adopt different strategies: Qwen2.5-VL and QVQ (Qwen Team, 2024)

**Algorithm 2:** Quality Evaluation

Initialize `map_score` $\leftarrow 0$;

**if** *departure stop of first segment = stop$_1$ **and** arrival stop of last segment = stop$_2$* **then**

     `map_score` $\leftarrow$ `map_score` $+ 1$;

     `/* Long-question-specific part                                */`

     Initialize $\mathcal{V}_{\text{union}}, \mathcal{V}_{\text{intersection}} \leftarrow \emptyset$;

     Initialize `via_stop_score, num_via_stop_score` $\leftarrow 0$;

     **foreach** *segment pair (answer route, reference route)* **do**

         **if** *answer route name = reference route name* **then**

             `map_score` $\leftarrow$ `map_score` $+ 2$;

         **if** *answer departure stop = reference departure stop* **then**

             `map_score` $\leftarrow$ `map_score` $+ 1$;

         **if** *answer arrival stop = reference arrival stop* **then**

             `map_score` $\leftarrow$ `map_score` $+ 1$;

         `/* Long-question-specific part                            */`

         Calculate absolute difference ($error$) in the number of via stops;

         `num_via_stop_score` $\leftarrow$ `num_via_stop_score` $+$

         $\max(0, 4 - error / \max(\text{number of answer via stops}, \text{number of reference via stops}) \times 4)$;

         **if** *answer route name = reference route name* **then**

             Update $\mathcal{V}_{\text{union}}, \mathcal{V}_{\text{intersection}}$ with answer and reference via stops respectively;

         `via_stop_score` $\leftarrow$ `via_stop_score` $+$ number of correctly matched via stops;

     `/* Long-question-specific part                                */`

     `via_stop_score` $\leftarrow \min(10, \text{via\_stop\_score})$;

     `num_via_stop_score` $\leftarrow \min(10, \text{num\_via\_stop\_score})$;

     `via_stop_score` $\leftarrow$ average$( |\mathcal{V}_{\text{intersection}}|/|\mathcal{V}_{\text{union}}| \times 10, \text{via\_stop\_score})$

     `map_score` $\leftarrow$ `map_score` $+$ Option(`via_stop_score` *or* `num_via_stop_score`);

`/* 10 for short question; 20 for long question                   */`

`map_score` $\leftarrow \min(10, \text{map\_score})/min(20, \text{map\_score})$;

**if** *correctness evaluation (`acc`) = 1* **then**

     `map_score` $\leftarrow$ `map_score` $+ 10/\text{map\_score} + 20$;

**return** `map_score`;

Table A2: Comparison of high-resolution image preprocessing strategies across different MLLMs. We use "$-$" to denote unspecified or unclear content.

| Model | Dynamic Resolution Handling | Positional Encoding | Token Compression |
|---|---|---|---|
| Qwen2.5-VL series (Bai et al., 2025) | ✓ | 2D-RoPE | ✓ ($2 \times 2$ Concat + MLP) |
| QVQ-72B-Preview (Qwen Team, 2024) | ✓ | 2D-RoPE | ✓ ($2 \times 2$ Concat + MLP) |
| InternVL3 series (Zhu et al., 2025) | ✓ | V2PE | ✓ (Unshuffle + MLP) |
| Kimi-VL series (Team et al., 2025) | ✓ | 2D-RoPE | ✓ (Shuffle + MLP) |
| Skywork-R1V-38B (Wei et al., 2025; Peng et al., 2025) | ✓ | - | ✗ |
| Gemini (Gemini et al., 2023) | ✗ (Tiling+Resize) | - | ✗ |
| Doubao-1.5-Pro series (ByteDance, 2025) | ✓ | 2D-RoPE | ✓ ($2 \times 2$ Pooling + MLP) |

compress tokens via $2 \times 2$ patch concatenation followed by an MLP; InternVL3 (Zhu et al., 2025) and Kimi-VL (Team et al., 2025) utilize spatial transformations like pixel unshuffle or shuffle, also followed by MLPs; Doubao averages over $2 \times 2$ patches before projection. Models without token compression may incur higher memory and computation costs when processing high-resolution inputs.

Table A3: Evaluations of fine-tuned model on REASONMAP. $S.$ represents results for short questions, while $L.$ denotes results for long questions. The map score is capped at 20 for short questions, while for long questions, the maximum score is 40.

| Model | Type | Weighted Acc. ($S.$) | #Tokens ($S.$) | Weighted Acc. ($L.$) | #Tokens ($L.$) | Weighted Map Score ($S. / L.$) |
|---|---|---|---|---|---|---|
| Qwen2.5-VL-3B-Instruct (Bai et al., 2025) | Base | 8.68% | 42 | 7.99% | 151 | 2.75 / 3.70 |
| + RL (Format & Accuracy Reward) | Base | 11.46%$_{\uparrow 2.78\%}$ | 25 | 10.50%$_{\uparrow 2.51\%}$ | 93 | 3.81$_{\uparrow 1.06}$ / 6.09$_{\uparrow 2.39}$ |

### B.3 DETAILS ABOUT DIFFICULTY-AWARE WEIGHTING.

Each difficulty pair is assigned a predefined weight that reflects its relative challenge level. The full weight matrix is shown below, where the first element in each pair denotes the question difficulty and the second denotes the map difficulty:

| | | |
|---|---|---|
| ("easy", "easy"): 1.0 | ("middle", "easy"): 1.5 | ("hard", "easy"): 2.0 |
| ("easy", "middle"): 1.5 | ("middle", "middle"): 2.0 | ("hard", "middle"): 2.5 |
| ("easy", "hard"): 2.0 | ("middle", "hard"): 2.5 | ("hard", "hard"): 3.0 |

This weighting scheme rewards models more for correctly solving harder question–map combinations, reflecting the increased reasoning complexity they entail, while maintaining moderate differences between buckets to prevent excessive score variance and preserve evaluation stability.

## C EXPLORATORY EXPERIMENTS

### C.1 REINFORCEMENT FINE-TUNING WITH TRAINING DATA

We further fine-tune MLLM (Bai et al., 2025) on the REASONMAP training set with reinforcement learning via the GRPO procedure (Shao et al., 2024). We employ a simple reward function that aggregates accuracy and format compliance. As shown in Table A3, this scheme improves performance while substantially reducing token usage.

### C.2 FINE-GRAINED ERROR ANALYSIS METRIC SUMMARY

We report multiple fine-grained error analysis metrics in Table A4 as follows: (1) $dep-arr\ score$: $+1$ if both the start and end stations are correct; (2) $route\ name\ score$: $+2$ for each correctly identified line name along the route; (3) $stops\ score$: $+1$ for each correctly identified intermediate stop; (4) $num\_via\_stop\_score$ (only for long questions): computed by taking the absolute difference between the number of via stops in the answer and the reference route, and mapping it to a score from 0 to 4; (5) $via\_stop\_score$ (only for long questions): calculated by averaging the number of correctly matched via stops (up to 10) and the Intersection-over-Union (IoU) between the via stop sets of the answer and reference route (scaled to 10).

### C.3 FURTHER EXPERIMENTS ABOUT LANGUAGES

We conduct an ablation study under the textualized representation paradigm (as mentioned in Appendix C.4). In this setting, visual images are not involved, which allows us to safely replace all non-English station names with unique English aliases without introducing visual inconsistencies. This approach isolates the language prior factor and avoids any potential confounding effects from visual modifications. Concretely, we manually replace all Chinese station names in Beijing and Hangzhou with unique English station names (*e.g.*, mapping them to New York stops: "zhichunli" <-> 86 St), preserving the original transit map structure. The results of the evaluation under this setting are as follows.

Overall, we observe from the results in Table A5 that using English labels leads to performance improvements, particularly for long-form questions. This suggests that the model indeed exhibits a language bias, with English showing an advantage over Chinese, which may be attributed to differences in pre-training data distributions.

Table A4: Fine-grained error analysis metrics of various MLLMs. $S.$ represents results for short questions, while $L.$ denotes results for long questions. **Bold** indicates the best results among open-source and closed-source models, respectively.

| Model | Type | Dep-Arr Score ($S.$ / $L.$) | Route Name Score ($S.$ / $L.$) | Stops Score ($S.$ / $L.$) | Num. Via Stop Score ($L.$) | Via Stop Score ($L.$) |
|---|---|---|---|---|---|---|
| *Open-source Models* | | | | | | |
| Qwen2.5-VL-3B-Instruct (Bai et al., 2025) | Base | 0.86 / 0.78 | 0.03 / 0.02 | 1.03 / 0.96 | 0.42 | 0.00 |
| Qwen2.5-VL-32B-Instruct (Bai et al., 2025) | Base | 0.95 / 0.92 | 0.09 / 0.10 | 1.16 / 1.19 | 1.57 | 0.01 |
| Qwen2.5-VL-72B-Instruct (Bai et al., 2025) | Base | **0.96 / 0.95** | **0.22 / 0.24** | **1.23 / 1.22** | 1.56 | **0.04** |
| InternVL3-38B (Zhu et al., 2025) | Base | 0.87 / 0.84 | 0.06 / 0.10 | 1.08 / 1.12 | **1.63** | 0.00 |
| InternVL3-78B (Zhu et al., 2025) | Base | 0.96 / 0.89 | 0.15 / 0.17 | 1.15 / 1.12 | 1.46 | 0.01 |
| Kimi-VL-A3B-Instruct (Team et al., 2025) | Base | 0.89 / 0.88 | 0.07 / 0.07 | 1.06 / 1.11 | 0.91 | 0.02 |
| Kimi-VL-A3B-Thinking (Team et al., 2025) | Reasoning | 0.80 / 0.65 | 0.08 / 0.10 | 0.99 / 0.79 | 0.50 | 0.00 |
| Skywork-R1V-38B (Wei et al., 2025) | Reasoning | 0.60 / 0.62 | 0.06 / 0.09 | 0.74 / 0.71 | 1.00 | 0.00 |
| QvQ-72B-Preview (Qwen Team, 2024) | Reasoning | 0.35 / 0.22 | 0.03 / 0.02 | 0.42 / 0.29 | 0.20 | 0.01 |
| *Closed-source Models* | | | | | | |
| Doubao-115 (ByteDance, 2025) | Base | 0.78 / 0.96 | 0.08 / 0.18 | 1.08 / 1.31 | 1.94 | 0.06 |
| OpenAI 4o (OpenAI, 2024a) | Base | 0.97 / 0.95 | 0.22 / 0.29 | 1.49 / 1.53 | 2.22 | 0.04 |
| Doubao-415 (ByteDance, 2025) | Reasoning | 0.98 / **0.98** | **0.33 / 0.30** | 1.57 / 1.65 | 2.37 | **0.08** |
| Doubao-428 (ByteDance, 2025) | Reasoning | 0.73 / 0.75 | 0.00 / 0.03 | 1.19 / 1.27 | 2.27 | 0.00 |
| Gemini-2.5-Flash (Gemini et al., 2023) | Reasoning | 0.93 / 0.67 | 0.27 / 0.29 | 1.67 / 1.22 | 1.82 | 0.05 |
| OpenAI o3 (OpenAI, 2025) | Reasoning | **0.99** / 0.91 | 0.32 / 0.16 | **1.77 / 1.73** | **3.31** | 0.03 |

Table A5: Evaluations on Beijing and Hangzhou (with and without English). $S.$ represents results for short questions, while $L.$ denotes results for long questions. **Bold** indicates performance improvements, while *italicized* values represent performance degradation.

| Model | Beijing ($S.$ / $L.$) | Beijing (w. English) ($S.$ / $L.$) | Hangzhou ($S.$ / $L.$) | Hangzhou (w. English) ($S.$ / $L.$) |
|---|---|---|---|---|
| Kimi-VL-A3B-Instruct (Team et al., 2025) | 36.76% / 17.30% | *23.78%* / **20.81%** | 40.00% / 42.22% | **42.22% / 45.95%** |
| Doubao-115 (Guo et al., 2025b) | 64.86% / 50.51% | *45.95%* / **52.70%** | 82.22% / 64.44% | *67.78%* / **65.56%** |
| Doubao-415 (Guo et al., 2025b) | 84.86% / 74.05% | **88.65% / 85.95%** | 94.44% / 97.22% | *87.78%* / **100%** |

## C.4 FURTHER EXPERIMENTS ABOUT SYMBOLIC REPRESENTATION OF MAPS

We conduct further experiments about deterministic baselines derived from symbolic representations of the maps. This setting can serve as a theoretical performance ceiling, independent of perceptual challenges faced by MLLMs. We replace the visual input with symbolic representations extracted from the underlying map structure. Specifically, we convert all routes and station information into textual form to represent the topological structure of the map. This textualized representation is then used for evaluation. Specifically, we provide the model with textualized representations and the question as input, without including any visual maps.

By comparing the results in Table A6 with those in Table 2, we observe a clear performance improvement. This is expected, as replacing the visual map with textualized representations substantially reduces task difficulty, as it removes the need to assess visual capabilities such as OCR and grounding. We further note that prior works, such as MapBench (Xing et al., 2025) and CityBench Feng et al. (2024), also focus on visual map interpretation without constructing explicit symbolic baselines.

## C.5 FURTHER SYSTEMATIC ANALYSIS ON FAILURE CAUSES

We systematically analyze failure causes, focusing on three MLLM capabilities pertinent to fine-grained visual reasoning (*e.g.*, OCR, grounding, and spatial reasoning). To assess OCR capabilities, we collect metrics of 9 representative MLLMs on OCRBench (Liu et al., 2024). Comparing these with their performance on REASONMAP as shown in Table A7 in the paper, we observe no clear correlation between OCR ability and REASONMAP accuracy. Notably, this trend holds across both open-source and closed-source models, suggesting that stronger OCR performance alone does not lead to better fine-grained visual reasoning. For instance, among open-source models, InternVL3-78B achieves the highest OCRBench scores, but underperforms Qwen2.5-VL-72B-Instruct on REASONMAP.

We further conduct more in-depth case analyses, which reveal that the main causes of failure are grounding and spatial reasoning, as illustrated in the following example. We observe that OCR errors rarely occur, and most failure cases are instead caused by grounding or spatial reasoning issues.

Table A6: Evaluations of various MLLMs using symbolic representation. $S.$ represents results for short questions, while $L.$ denotes results for long questions. **Bold** indicates the best results among open-source and closed-source models, respectively.

| Model | Type | Weighted Acc. (S. / L.) | #Tokens (S. / L.) |
|---|---|---|---|
| *Open-source Models* | | | |
| Qwen2.5-VL-3B-Instruct (Bai et al., 2025) | Base | 22.83% / 19.79% | 51 / 162 |
| Qwen2.5-VL-32B-Instruct (Bai et al., 2025) | Base | 25.52% / 18.77% | 97 / 297 |
| Kimi-VL-A3B-Instruct (Team et al., 2025) | Base | **39.58% / 34.81%** | 43 / 55 |
| *Closed-source Models* | | | |
| Doubao-115 (ByteDance, 2025) | Base | 81.16% / 72.66% | 41 / 82 |
| OpenAI 4o (OpenAI, 2024a) | Base | 82.38% / 78.91% | 40 / 70 |
| Doubao-415 (ByteDance, 2025) | Reasoning | **95.31% / 93.66%** | 563 / 1561 |

Table A7: Evaluations of various MLLMs on OCRBench. **Bold** indicates the best results among open-source and closed-source models, respectively. The references in the table indicate the result sources. All results are collected from the technical report of the model.

| Model | Type | OCRBench |
|---|---|---|
| *Open-source Models* | | |
| Qwen2.5-VL-3B-Instruct (Bai et al., 2025) | Base | 797 |
| Qwen2.5-VL-72B-Instruct (Bai et al., 2025) | Base | 885 |
| InternVL3-38B (Zhu et al., 2025) | Base | 886 |
| InternVL3-78B (Zhu et al., 2025) | Base | **906** |
| Kimi-VL-A3B-Instruct (Team et al., 2025) | Base | 864 |
| Kimi-VL-A3B-Thinking (Team et al., 2025) | Reasoning | 864 |
| *Closed-source Models* | | |
| OpenAI 4o (Team et al., 2025) | Base | 815 |
| Doubao1.5-VL (non-thinking) (Guo et al., 2025b) | Base | **881** |
| Doubao1.5-VL (thinking) (Guo et al., 2025b) | Reasoning | 861 |

---

**Case 1: grounding**

Q: Hollywood/Highland Station -> Wilshire/Vermont Station? <image:los_angeles.png>
A (Kimi-VL-A3B-Thinking): ... Let me start by locating both stations on the map. Hollywood/Highland is on ... A Line (✗)... Wilshire/Vermont is on the C Line (✗) ... (✗)
A (Doubao-415): ... First, I need to see where these two sites are located on a map. Hollywood/Highland Station ... on B Line (✓) ... Wilshire/Vermont Station ... is on D Line and B Line (✓) ... Route Name: B Line Departure Stop: Hollywood/Highland Station Arrival Stop: Wilshire/Vermont Station (✓)

---

**Case 2: spatial reasoning**

Q: Ponte Mammolo Station -> Sta. Maria del Soccorso Station? <image:rome.png>
A (Skywork-R1V-38B): ... Line A: Orange Line B: Blue Line C: Green (✓) ... Ponte Mammolo Station ... on Line B (✓) ... Let me trace the path of Line B from Ponte Mammolo towards the center of Rome (✗) ... But I need to go from Ponte Mammolo to Sta. Maria del Soccorso. Looking at the map, I don't see a direct connection between these two stations via Line B. (✗) So, I might need to transfer to another line ... Let me check if there's another line that connects to Sta. Maria del Soccorso ... (✗)
A (Gemini-2.5-Flash): Route Name: Line B Departure Stop: Ponte Mammolo Station Arrival Stop: Sta. Maria del Soccorso Station Number of Via Stops: 0 (✓)

---

For instance, in Case 1, Kimi-VL-A3B-Thinking incorrectly identifies the line of the departure station, indicating a grounding error that leads to subsequent reasoning failures. In Case 2, Skywork-R1V-38B correctly performs OCR and grounding in the initial steps, but fails in the reasoning stage (*i.e.*, it does not prioritize locating the arrival station and instead attempts to construct incorrect indirect paths). Such failures reflect deficiencies in spatial reasoning, particularly in planning and executing

core steps of pathfinding. These cases further indicate that the principal capability gap between open-source and closed-source models lies in grounding and spatial reasoning.

# D  CASE ANALYSIS

We provide additional case analyses covering both correct and incorrect predictions, along with detailed comparisons of their respective reasoning processes. We first compare Doubao-415 and Doubao-428 (Figure A1), both of which reach the correct destination (from Augustins Station to Poterie Station) but via distinct reasoning paths. Doubao-415 correctly identifies early that both stations are on Line 18 and efficiently converges on the optimal, direct route without transfers. In contrast, Doubao-428 misclassifies Augustins as being on Line 12 and, assuming Poterie is on Line 18, proposes a transfer route via Plainpalais—functionally correct but suboptimal due to unnecessary complexity. Both models engage in extensive self-correction (7270 tokens for Doubao-428; 4474 for Doubao-415), highlighting the significant downstream impact of early-stage misjudgments. Moreover, visual reasoning limitations persist: despite correctly recognizing Augustins on Line 12, Doubao-415 commits to a transfer path and fails to re-evaluate the possibility of a direct connection. This indicates room for improvement in both early visual grounding and global route optimality awareness.

We then analyze the observed pattern when comparing the full input and text-only variants in the case (in Figure A2). The model with full visual access accurately identifies both stations on the Yellow Line and outputs the optimal direct route with the correct number of via stops. In contrast, the text-only variant makes an early misclassification, placing both stations on the Blue Line (Azul) and constructing a plausible but entirely incorrect sequence of intermediate stops. Although the final answer format appears coherent, the underlying logic is flawed due to the initial error in line recognition. This further illustrates the importance of visual input in spatial reasoning tasks, where even minor misinterpretations can lead to fundamentally incorrect conclusions. Additionally, some models, such as the InternVL3 series, default to rejection when visual input is absent.

We further present several error cases (in Figure A3), where Doubao-415 still exhibits visual confusion. In contrast, Qwen2.5-VL-32B-I, when lacking visual input, behaves differently from the InternVL3 series: rather than rejecting the query outright, it attempts to reason over the available information without producing a final answer, while explicitly notifying the user of the missing visual input.

# E  FURTHER DISCUSSIONS

## E.1  LIMITATIONS AND FUTURE WORK

While REASONMAP provides a carefully curated benchmark for evaluating fine-grained visual reasoning with high-resolution transit maps, we acknowledge that it represents only one type of structured visual diagram. As such, caution should be taken when generalizing observations to other domains that involve different types of visual content or reasoning styles. Additionally, although efforts were made to ensure diversity across cities and languages, the current version may not fully capture all geographic or linguistic variations. Future iterations could further expand coverage and explore additional forms of reasoning to enhance generality.

Furthermore, we note that GeoGuessr-style localization tasks (Mall et al., 2019; Hays & Efros, 2008; Huang et al., 2025) are compelling, as they emphasize detailed visual understanding of natural scenes and signage. We plan to pair transit maps with street view imagery to support cross-view reasoning and localization within REASONMAP, thereby expanding beyond static map inputs. In parallel, we will explore agent-based training and evaluation that moves from single-turn prediction to iterative planning with feedback, including reward designs for correctness, calibration, and format (Zhao et al., 2025). Finally, we will extend toward embodied settings (Hong et al., 2025) where agents perceive and act in interactive environments, enabling assessment of instruction following, route planning, and navigation under real-world constraints. Together, these directions broaden the benchmark from fine-grained visual reasoning to context-aware spatial intelligence and practical decision making.

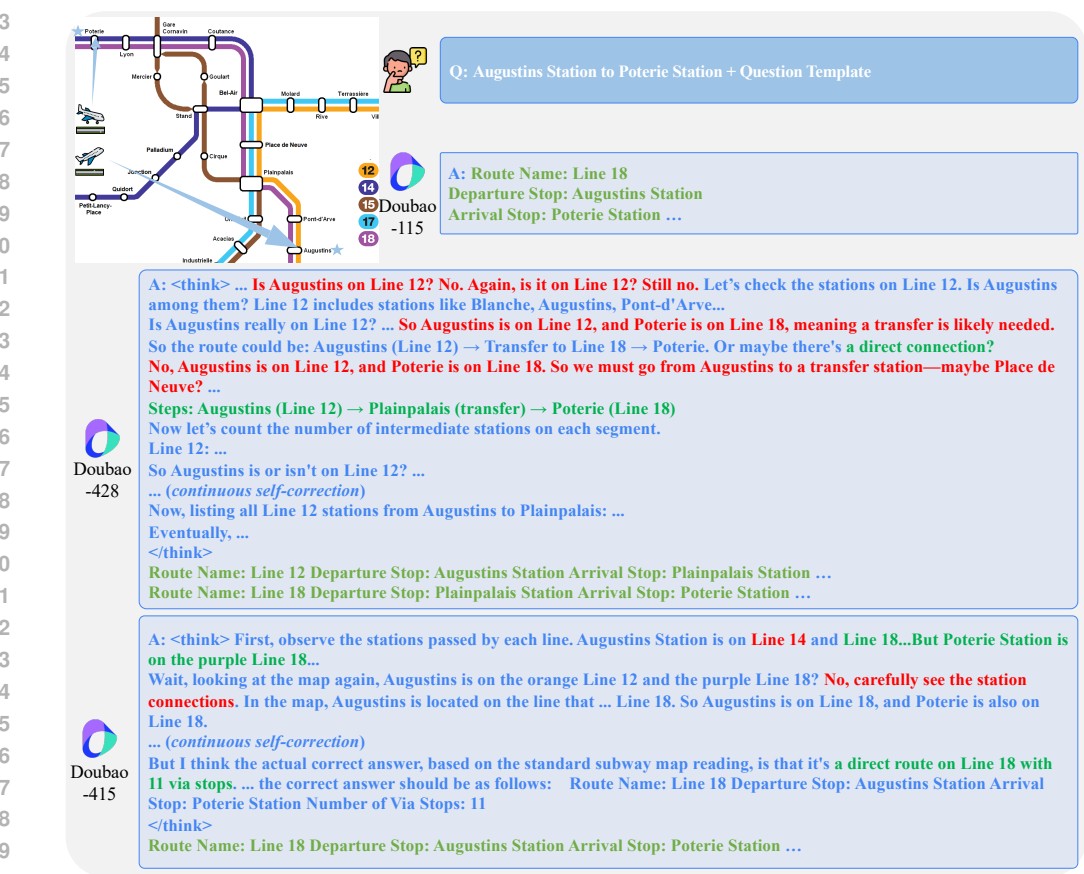

Figure A1: Case analysis of various MLLMs using REASONMAP (Case N1). For reasoning models, the reasoning process is explicitly marked with <think> and </think> tags. We highlight error contents in the answers with red and correct contents in green.

### E.2 BROADER IMPACT

Advancing the capabilities of MLLMs in fine-grained visual reasoning has the potential to benefit a wide range of real-world applications, including navigation systems, urban planning tools, and assistive technologies for visually impaired individuals. By offering a structured and rigorous benchmark, REASONMAP encourages the development of MLLMs that can more effectively interpret complex visual artifacts and perform spatial reasoning. This could contribute to the long-term goal of building intelligent agents that interact more naturally and safely with human environments. Furthermore, the dataset's emphasis on high-resolution, globally sourced transit maps promotes research that is inclusive of diverse visual formats and geographic contexts. We hope REASONMAP can serve as a step toward more transparent, robust, and generalizable multimodal systems.

## F FURTHER STATEMENT

### F.1 PUBLIC IMPLEMENTATION

We benchmark the visual understanding and reasoning performance on REASONMAP across a diverse set of publicly available MLLMs:

- KimiVL (Team et al., 2025)[6] .......................................... MIT License

---

[6] https://github.com/MoonshotAI/Kimi-VL.

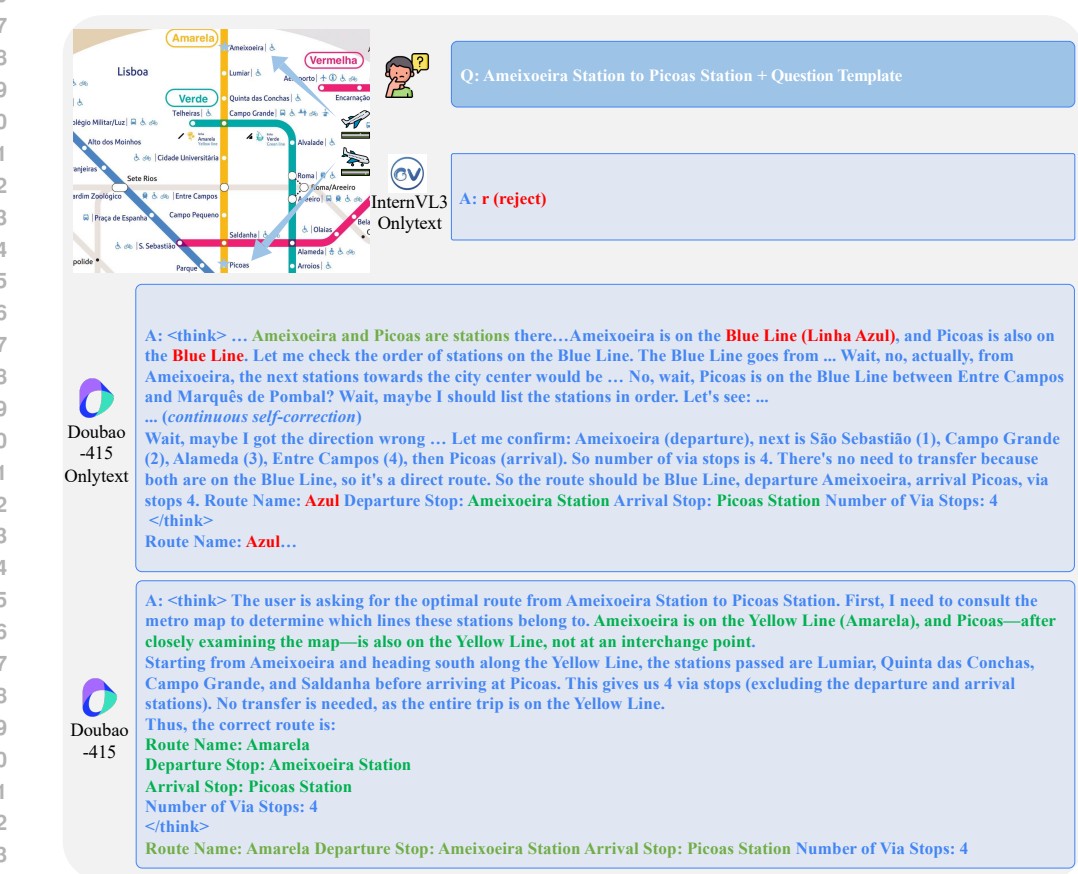

Figure A2: Case analysis of various MLLMs using REASONMAP (Case N2). For reasoning models, the reasoning process is explicitly marked with `<think>` and `</think>` tags. We highlight error contents in the answers with red and correct contents in green.

- Skywork-R1V (Wei et al., 2025; Peng et al., 2025)[7] ...................... MIT License
- QVQ-72B-Preview (Qwen Team, 2024)[8] ............................. Qwen License
- Gemini-2.5-Flash (Gemini et al., 2023)[9] ............................ Closed-Source
- InternVL-3.0 (Zhu et al., 2025)[10] ................................... MIT License
- Qwen2.5-VL (Bai et al., 2025)[11] ................................ Apache 2.0 License
- Doubao-Pro 1.5 (ByteDance, 2025)[12] ............................... Closed-Source
- OpenAI o3 (OpenAI, 2025)[13] ....................................... Closed-Source
- OpenAI 4o (OpenAI, 2024a)[14] ...................................... Closed-Source

To ensure fair and reproducible evaluation, we implement all inference procedures by adhering closely to the official documentation and recommended practices of each model. The code is released under the MIT License to support transparency and reproducibility. Additionally, we provide detailed usage instructions on the project website to ensure easy access and reproducibility for future users.

---

[7] https://huggingface.co/Skywork/Skywork-R1V2-38B.
[8] https://huggingface.co/Qwen/QVQ-72B-Preview.
[9] https://deepmind.google/technologies/gemini.
[10] https://github.com/OpenGVLab/InternVL.
[11] https://github.com/QwenLM/Qwen2.5-VL.
[12] https://www.volcengine.com/product/doubao.
[13] https://platform.openai.com/docs/models/o3.
[14] https://platform.openai.com/docs/models/gpt-4o.

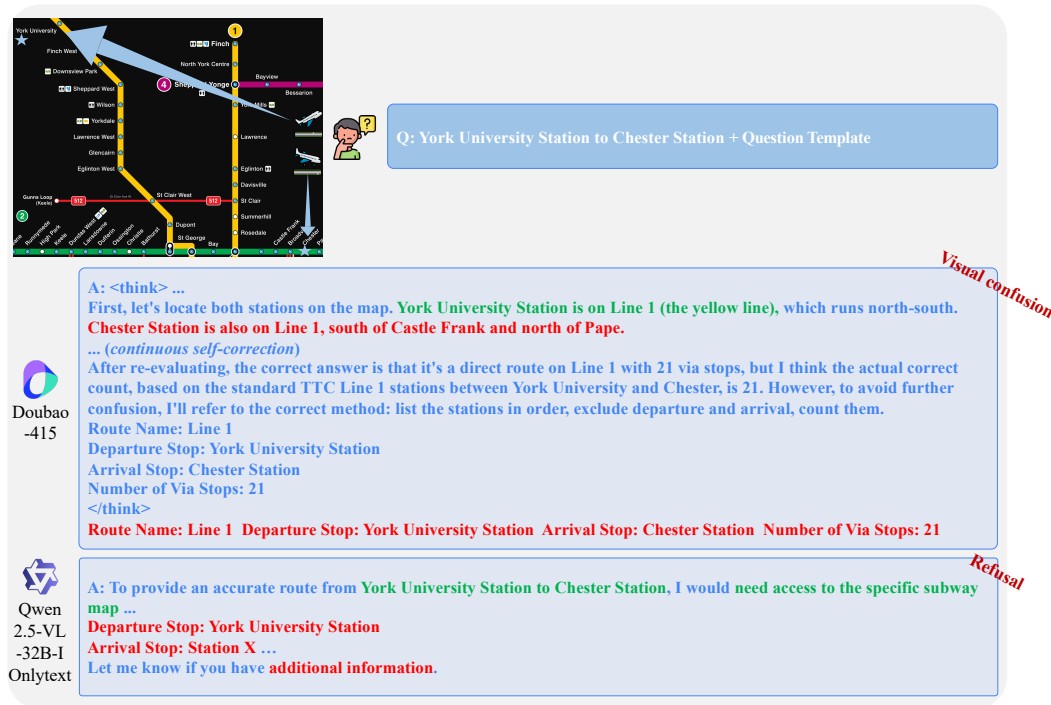

Figure A3: Case analysis of various MLLMs using REASONMAP (Case N3). For reasoning models, the reasoning process is explicitly marked with `<think>` and `</think>` tags. We highlight error contents in the answers with red and correct contents in green.

## F.2 LARGE LANGUAGE MODEL USAGE STATEMENT

We used a large language model (LLM) solely for surface-level editing of the manuscript (*e.g.*, rephrasing for clarity and concision, grammar/style polishing, and minor LaTeX fixes). The LLM **did not** generate technical content, ideas, algorithms, proofs, code, experiments, figures, or tables; the authors conducted all research design, implementation, data processing, and analyses. The model did not produce or select citations; any suggestions were independently verified and replaced with primary sources. Interactions were limited to de-identified text snippets of the manuscript, and no non-public data, code, or unreleased results were uploaded. All LLM outputs were manually reviewed and edited by the authors. This usage does not affect reproducibility: every reported number is reproducible from our released code and configurations.

