# OpenReview forum: "Can MLLMs Guide Me Home? A Benchmark Study on Fine-grained Visual Reasoning from Transit Maps"
_ICLR.cc/2026/Conference — ICLR 2026 Conference Withdrawn Submission_

### Official Review · Reviewer_Q15t · 2025-10-26

**Soundness:** 3
**Presentation:** 2
**Contribution:** 3
**Rating:** 4
**Confidence:** 5

**Summary:**

This paper introduces REASONMAP, a benchmark to evaluate the visual reasoning of MLLMs using high-resolution transit maps. The task requires models to find routes between stations. The authors propose a two-level evaluation metric to assess both the correctness and quality of the generated routes and present results from 15 MLLMs.

**Strengths:**

1. The paper tackles a valuable and challenging problem: testing MLLM capabilities on dense, high-resolution, practical-use diagrams, which is a clear gap in current benchmarks.
2. The two-level evaluation framework, which separates route correctness (is it valid?) from route quality (is it optimal?), is a strong contribution. This is a nuanced approach that is well-suited for this pathfinding task.

**Weaknesses:**

1. The dataset is too small for objective evaluation. 30 cities and 1008 QA pairs are insufficient to draw general conclusions. Furthermore, these 30 cities may be popular, meaning their transit systems are likely part of the MLLMs' pre-training data. This contaminates the results, making it unclear if models are demonstrating visual reasoning or recalling prior knowledge.

2. Why was a 20:15:5 (easy:medium:hard) ratio chosen for question difficulty? The paper claims this is for "balance," but a 1:1:1 split would be balanced. The rationale is not provided.

3. How were maps split into easy, medium, and hard? The paper mentions this was done manually and lists proxies in the appendix, but the actual logic or thresholds used for this assignment are missing.

4. The paper mentions "automatic checks and manual adjustments" for quality control but provides no details on what these checks were or how they were performed.

6. The dataset is too small to be split. A test set of 312 samples across 11 cities is insufficient and likely biased. How were these 11 cities chosen? How many of them fall into the multilingual category to test robustness?

7. The difficulty categories (map difficulty x question difficulty) combined with a separate difficulty-weighting scheme make the results convoluted and hard to interpret. A single, holistic difficulty score for each QA pair, combining all factors, would be simpler and more effective.

8. The analysis section is shallow. The main takeaway from the difficulty breakdown (Fig. 3) is that "performance degrades as task complexity increases," which is an obvious conclusion. The paper observes trends (e.g., base vs. reasoning models) but fails to provide a deep diagnosis why these trends occur, beyond citing prior work.

9. The paper incorrectly uses "fine-grained visual understanding" and "spatial reasoning" interchangeably. This benchmark clearly tests reasoning (planning a path). However, it's less of a pure test of visual understanding, which would be more about perception (e.g., "find this station," "trace this line"). This benchmark has potential as a diagnostic tool, but its claim to be a definitive measure of fine-grained visual understanding is not fully supported.

**Questions:**

1. What is the specific rationale for the 20:15:5 (easy:medium:hard) question ratio, and why is this "balanced"?
2. What exact logic and thresholds were used to manually categorize the 30 maps into difficulty levels?
3. What specific "automatic checks" and "manual adjustment" protocols were used for quality control?
4. How many of the 312 test samples are non-English, and why is this sufficient to evaluate multilingual performance?

---

### Official Review · Reviewer_foUs · 2025-10-26

**Soundness:** 2
**Presentation:** 2
**Contribution:** 2
**Rating:** 2
**Confidence:** 4

**Summary:**

This paper introduces REASONMAP, a new benchmark dataset designed to evaluate the fine-grained visual understanding and spatial reasoning capabilities of Multimodal Large Language Models (MLLMs). The dataset consists of 1,008 question-answer pairs based on high-resolution transit maps from 30 cities. The authors also propose a two-level evaluation framework to assess both the correctness and the quality of the models' generated routes. A key finding from their experiments on 15 MLLMs is a counterintuitive performance gap, where open-source base models outperform their reasoning-focused counterparts, while the opposite is true for closed-source models.

**Strengths:**

1. **Novel and Important Problem:** The paper addresses a critical and timely gap in MLLM evaluation. The task of reasoning over complex, information-dense transit maps pushes beyond simple OCR or captioning tasks, requiring genuine fine-grained visual understanding and multi-step spatial reasoning, which is a significant step forward for multimodal benchmarks.

2. **Strong Benchmark Validation:** The inclusion of an experiment where visual inputs are masked is a major strength. This ablation effectively validates that the benchmark tests visual perception and reasoning rather than relying solely on the models' pre-existing parametric knowledge, confirming the dataset's value to the community.

**Weaknesses:**

The paper's primary weaknesses stem from the limited scale of its evaluation, a perceived arbitrariness in its evaluation framework, and a lack of deeply grounded analysis for some of its key findings.

---

### Small and Potentially Biased Evaluation Set
A significant limitation is the size of the test set. The entire dataset contains just over 1,000 question-answer pairs, and the evaluation is conducted on a small subset of only 312 examples. While the authors reserve the rest for future training use, this small evaluation set raises concerns about the statistical significance and generalizability of the findings. With such a limited number of data points, the results may be susceptible to noise and may not be representative of the models' true capabilities across a wider range of maps and questions. This small scale is a critical weakness for a paper introducing a new benchmark.

---

### Subjectivity and Lack of Rigor in the Evaluation Framework
Several core components of the evaluation methodology feel arbitrary and are not sufficiently justified, which undermines the reliability of the results. A more principled and transparent approach is needed.

**Map Difficulty:** The difficulty level of maps ("easy," "medium," "hard") is assigned manually without a clear, replicable metric. While the appendix mentions proxies like line and transfer counts, the paper would be much stronger if it defined a concrete heuristic or a quantitative formula used for this categorization. Without it, the classification appears subjective.

**Arbitrary Scoring and Weighting:** The point system for the "map score" and the "difficulty-aware" weighting scheme both seem ad-hoc. For instance, the rationale for assigning specific point values (e.g., two points for a route name vs. one for an endpoint) is not explained. Similarly, the 3x3 weight matrix for combining map and question difficulty adds a layer of complexity that feels convoluted and whimsical rather than methodologically sound. These arbitrary choices make the final performance scores difficult to interpret and trust.

---

### Speculative Analysis and Unanswered Questions
The paper presents interesting results but often stops short of providing a rigorous, evidence-based analysis, leaving key questions unanswered.

**Unsupported Claims on Model Performance:** The paper's explanation for why open-source "thinking" models underperform their base versions is presented as speculation rather than a conclusion drawn from rigorous analysis. The authors suggest this may be due to RL-induced biases, but this claim is not substantiated with targeted experiments. A deeper analysis, such as examining the models' reasoning traces or conducting specific ablations, is needed to provide a more grounded explanation for this surprising and important finding.

**Confounding Experimental Variables:** The decision to cap the maximum output tokens at 2,048 for open-source models is a potential confounder that is not adequately addressed. It is plausible that this limit prematurely terminated the "chain-of-thought" process for the reasoning-centric models, thereby artificially depressing their performance. The paper needs to clarify the rationale for this choice and explain why a similar constraint was not applied or reported for the closed-source models, as this is crucial for a fair comparison.

---

### Clarity and Readability
The paper is dense and can be difficult to follow at times. Simplifying the language and improving the narrative flow would help make the contributions more accessible and engaging for a broader audience.

**Questions:**

1. Could you provide a specific, quantifiable heuristic or formula that was used to manually classify the maps into "easy," "medium," and "hard" difficulty levels?

2. What is the justification for the point values assigned in the "map score" (e.g., why is a correct route name worth two points while an endpoint is worth one)?

3. Could you elaborate on the methodology and reasoning behind the 3x3 "difficulty-aware" weighting matrix used in your evaluation?

4. Regarding the 2,048 maximum token limit for open-source models, could this have prematurely cut off the "chain-of-thought" reasoning process? Why was a similar constraint not reported for the closed-source models?

5. Given that the evaluation was conducted on a small test set of 312 examples, have you performed any statistical tests (e.g., bootstrapping) to confirm the stability and significance of your findings?

---

### Official Review · Reviewer_YMqW · 2025-10-30

**Soundness:** 2
**Presentation:** 3
**Contribution:** 3
**Rating:** 6
**Confidence:** 4

**Summary:**

The paper introduces REASONMAP, a benchmark dataset for evaluating the fine-grained visual understanding and spatial reasoning capabilities of MLLMs. The core task requires MLLMs to perform pathfinding by interpreting high-resolution transit maps. The dataset comprises 1,008 question-answer pairs derived from public-domain maps of 30 cities across 13 countries. The authors employ a semi-automated data creation pipeline: MLLMs are first used to extract transit line and stop names into a structured JSON format, which is then manually verified. Ground-truth routes between two randomly selected stops are generated using map service APIs (Google Maps and Gaode Maps). The tasks are structured into three templates: one short question (identifying the route) and two long question variants (additionally identifying either the number of intermediate stops or the names of intermediate stops). The authors manually annotated map difficulty and used the number of transfers in the ground-truth route to programmatically set question difficulty. The benchmark is evaluated on a sample test set using 15 MLLMs. The primary experimental finding is a significant performance divergence: closed-source reasoning models outperform their base counterparts, whereas the opposite is true for open-source models. Ablation studies confirm that while models use visual information, they also rely heavily on prior knowledge.

**Strengths:**

1. The paper details a semi-automated data creation pipeline. Great detail is paid to ensure a high-quality dataset that is diverse in map style and language.
2. The inclusion of text-only and language-replacement experiments (in Appendix) provides valuable insights. These ablations effectively help disentangle the model's reliance on visual grounding versus its pre-existing world knowledge, strengthening the paper's conclusions.
3. The paper is generally well-written and easy to follow. The overview figures effectively illustrate the dataset's components and the data creation pipeline.

**Weaknesses:**

1. The paper's core motivation is unconvincing and missed the "why" for this specific task. Given that navigation is a well-solved problem via APIs (which the authors use for ground truth), it is unclear why an MLLM must perform this end-to-end visual reasoning. The task could be more realistically framed as a test of tool-use. The paper does not justify why this capability is intrinsically valuable for a standalone MLLM.
2. The evaluation relies on strict string matching for correctness. This is overly brittle and penalizes models for trivial, semantically irrelevant formatting deviations (e.g., "Line 1" vs. "1"), stylistic variations, or different language choices (on multi-lingual maps). A more robust semantic evaluation like using an MLLM judge would be more appropriate for assessing true reasoning.
3. The paper contains several magic numbers without justification. The 20:15:5 (easy:medium:hard) question difficulty ratio is arbitrary. The scoring algorithm seems hand-tuned. These choices lack a principled foundation and affect the final reported scores.
4. The font size in all tables is very small. Additionally, Figure 3 should include data labels on the bars to convey the precise accuracy numbers.

**Questions:**

1. For a given stop-pair, why was only one of the two long templates randomly assigned? Why not test both templates for the same route to measure the model's robustness to different formatting requirements?
2. How were multilingual maps handled during data creation and evaluation? If a map has stop names in two languages, which language was chosen for the ground truth, and how are answers in the other valid language scored by your strict evaluation?
3. How is manual annotation done for different languages?

---

### Official Review · Reviewer_vmxC · 2025-10-31

**Soundness:** 3
**Presentation:** 3
**Contribution:** 3
**Rating:** 4
**Confidence:** 4

**Summary:**

This paper presents REASONMAP, a benchmark for evaluating fine-grained spatial reasoning in multimodal large language models using high-resolution transit maps. The dataset includes about 1,000 human-verified QA pairs from 30 cities and introduces a two-level evaluation that measures both answer correctness and route quality (map score). The authors benchmark 15 MLLMs across open-source and closed-source variants and find that open-source reasoning models often perform worse than their base versions, while closed-source reasoning models perform better. Additional experiments with masked visual input show that most models rely heavily on textual priors instead of true visual grounding.

**Strengths:**

1. Originality: Using transit maps as a testbed for multimodal reasoning is creative and well-motivated. Prior datasets such as MathVerse, VisuLogic, and CityBench rely on synthetic or narrow tasks, while this one focuses on structured, real-world visuals that naturally require spatial understanding.


2. Quality: The dataset pipeline is solid, combining automated extraction with human verification and balanced difficulty design. The two-level evaluation (accuracy and map score) provides a more nuanced picture of reasoning quality than simple accuracy metrics.


3. Significance: The findings reveal a real limitation in current MLLMs: their weak visual grounding even on structured and information-rich inputs. The contrast between open and closed models is also interesting.

**Weaknesses:**

1. The dataset is relatively small (around 1k QA pairs), which makes it hard to draw statistically strong conclusions. Reporting variation across random subsets or expanding the dataset would help.


2. The map-score metric feels somewhat heuristic. There is no validation showing that it correlates with human judgment of route plausibility or quality.


3. The explanation for why reasoning models underperform is not fully convincing. It could also be due to token length limits or strict output formatting rather than just RL-based training bias.

**Questions:**

1. Have you checked if the map-score metric aligns with human judgments, for example through correlation on a small subset?
2. Could the weaker performance of reasoning models be due to longer outputs or token truncation?
3. Did you try cross-city generalization, such as training on some cities and testing on unseen ones?
4. Have you measured how much performance changes when using lower-resolution maps to test fine-grained perception?

---

### Note · Authors · 2025-11-14

I have read and agree with the venue's withdrawal policy on behalf of myself and my co-authors.